# Composite Fibers Based on Polycaprolactone and Calcium Magnesium Silicate Powders for Tissue Engineering Applications

**DOI:** 10.3390/polym14214611

**Published:** 2022-10-30

**Authors:** Cristina Busuioc, Andrada-Elena Alecu, Claudiu-Constantin Costea, Mihaela Beregoi, Mihaela Bacalum, Mina Raileanu, Sorin-Ion Jinga, Iuliana-Mihaela Deleanu

**Affiliations:** 1Department of Science and Engineering of Oxide Materials and Nanomaterials, Faculty of Chemical Engineering and Biotechnologies, University Politehnica of Bucharest, RO-060042 Bucharest, Romania; 2National Institute of Materials Physics, RO-077125 Magurele, Romania; 3National Institute of Physics and Nuclear Engineering, RO-077125 Magurele, Romania

**Keywords:** polycaprolactone, diopside, akermanite, merwinite, electrospinning, bone scaffolds

## Abstract

The present work reports the synthesis and characterization of polycaprolactone fibers loaded with particulate calcium magnesium silicates, to form composite materials with bioresorbable and bioactive properties. The inorganic powders were achieved through a sol–gel method, starting from the compositions of diopside, akermanite, and merwinite, three mineral phases with suitable features for the field of hard tissue engineering. The fibrous composites were fabricated by electrospinning polymeric solutions with a content of 16% polycaprolactone and 5 or 10% inorganic powder. The physico-chemical evaluation from compositional and morphological points of view was followed by the biological assessment of powder bioactivity and scaffold biocompatibility. SEM investigation highlighted a significant reduction in fiber diameter, from around 3 μm to less than 100 nm after the loading stage, while EDX and FTIR spectra confirmed the existence of embedded mineral entities. The silicate phases were found be highly bioactive after 4 weeks of immersion in SBF, enriching the potential of the polymeric host that provides only biocompatibility and bioresorbability. Moreover, the cellular tests indicated a slight decrease in cell viability over the short-term, a compromise that can be accepted if the overall benefits of such multifunctional composites are considered.

## 1. Introduction

Tissue engineering (TE) means the development of biological substitutes to “restore, maintain, or improve tissue function” [1]. One of the strategies of this interdisciplinary field is represented by the design of suitable materials to be implemented as scaffolds. Depending on characteristics, a scaffold can offer temporary cell support, space filling, or be utilized as releasing matrix of active molecules [2]. Numerous biocompatible materials have been researched and clinically studied so far, including metals, ceramics, polymers, and composites; however, choosing the most suitable microenvironment to stimulate cellular adhesion, growth, and differentiation is still a challenge [3,4].

Polymer–bioceramic composites, which result from combining two or more distinct phases, can be successfully used as heterogenous functional matrices with the ability to mimic the natural extracellular matrix (ECM), and are currently intensively researched [5]. Depending on their composition, these kinds of materials can combine the advantages of each constituent to offer a biodegradability rate compatible with morphogenesis rate, biocompatibility, non-toxicity, non-immunogenicity, and adequate morphological and mechanical properties [2,5,6,7,8]. In the simplest formulation, a polymer–bioceramic composite is fabricated employing a polymer, to offer flexibility, and a bioceramic, to provide proper mechanical properties; this is similar to bone ECM, which is composed of approximately 35% organic matrix (collagen fibrils) and 65% mineral reinforcement (crystallized calcium phosphate) [2,5].

Among bioceramics, calcium magnesium silicates-based systems play an important role in TE due to their proven biocompatibility, excellent mechanical properties, and high versatility, as their chemical composition and structure can be easily redesigned to respond to different clinical conditions [9]. During the degradation process, these systems release bioactive ions at a controlled rate, with major benefits for bone formation and the human body. Specifically, Ca^2+^, as the main element of bone tissue, is vital for osteogenesis, supporting cell growth and adapting cellular responses to bioceramics; Si^2+^ inhibits osteoporosis and stimulates the metabolic pathways of bone calcification; and Mg^2+^ modulates the degradation rate and mechanical strength of calcium silicates [6,10,11,12,13,14,15]. Furthermore, it has been found that Si^2+^ and Mg^2+^ are more important than Ca^2+^ in the process of cell differentiation [10]. So, in other words, each element has its role, but the properties of the resulting biomaterials are directly related to the ions’ relative concentration [9,14].

Considering all the above, it is easy to acknowledge ternary silicate bioceramics, such as diopside (D), akermanite (A), or merwinite (M), as frequently used materials in bone regeneration applications. Numerous studies previously investigated their properties, including the capacity to inhibit microbial growth, which is particularly critical for bone matrix [9,16,17], and concluded their applicability for TE as pure structures [6,10,18,19,20], or doped for improved bioactivity [21,22,23,24]. Calcium magnesium silicates have been produced in the form of particles (subsequently employed as fillers or decorations), thin or thick coatings, as well as porous scaffolds; most researchers have reported suitable mechanical properties, high bioactivity, and lack of cytotoxicity. CaO–MgO–SiO_2_-based bioceramics can be fabricated by wet or dry techniques: coprecipitation, sol–gel, combustion, spray pyrolysis, fusion process, solid-state sintering, etc. [6,13,15,25,26,27]. In our work, we chose the sol–gel method; although this requires rather expensive reagents and longer reaction times, it allows good control of the composition and structural characteristics of the obtained powders [13,25].

With respect to the main classes of polymers used in TE, natural and synthetic, the latter category offers the advantage of controlled composition and structure, tunable biodegradability, and possibility of functionalization. Polycaprolactone (PCL), as a synthetic material that enables good processability, precise control over degradation, molecular weight, or hydrophobicity [1], is often employed in nanofibrillar form in TE, drug delivery and wound healing [28,29,30,31,32,33,34,35,36]. It is approved by FDA and its biodegradability, bioresorbability, biocompatibility, and hydrophilicity can be modelled by the addition of inorganic additives [37]. Solution blow spinning, centrifugal spinning, electrospinning, and pressurized gyration have been studied to obtain PCL nanofibers [38]. Each method has advantages and drawbacks. Electrospinning, limited by its low yields relative to the industrial scale, is still one of the most used as it is a cost-effective, easy-to-apply technique in TE [4,39,40].

Thus, in this work, three types of mineral powders, differing in terms of calcium content, were loaded on polycaprolactone fibers by introducing them in the precursor electrospinning solutions. The novelty of this work resides in the achievement of new composites based on calcium magnesium silicates and polycaprolactone, namely the combination of the bioactive properties of the first with the bioresorbability of the second. Since few studies are available on this topic, their morphological and biological characteristics were investigated, and correlations were established with the compositional and processing parameters.

## 2. Materials and Methods

### 2.1. Powder Synthesis

The inorganic powders were synthesized by a sol–gel method and characterized from compositional, structural, and morphological points of view [41]. The sol–gel method is a wet-chemistry approach that involves the conversion of salt or alkoxide-type precursors first in a solution or colloidal suspension and then in a gel with high viscosity by hydrolysis and polycondensation/polymerization processes; such a gel consists of an extended network built on bridging oxygen and with high amounts of liquid phase embedded [42]. Briefly, three different powders were processed, having as a starting point the oxide compositions of diopside (CaMgSi_2_O_6_, D), akermanite (Ca_2_MgSi_2_O_7_, A), and merwinite (Ca_3_MgSi_2_O_8_, M); these were thermally treated at 1000 or 1300 °C. The pulverulent samples employed as mineral loading consist of consolidated blocks with a high percentage of open porosity; the first composition (D) contains diopside as leading crystalline compound, the second (A) is a balanced mixture of diopside, akermanite, and merwinite, and the third (M) has a majority of dicalcium silicate [41].

### 2.2. Composites Preparation

The composite fibers were fabricated through an electrospinning technique using chloroform (CF, CHCl_3_, ≥99%, Sigma-Aldich, St. Louis, MO, USA) and N,N-dimethylformamide (DMF, C_3_H_7_NO, 99.8%, Sigma-Aldich) as solvents, polycaprolactone (PCL, (C_6_H_10_O_2_)_n_, *Mw* = 80,000 g/mol, Sigma-Aldich) as the polymeric phase, and the previously described powders as the inorganic component (D, A, M). The CF:DMF solvent ratio was set at 4:1. Electrospinning uses electrical forces to shape a jet of polymeric solution with adequate rheological properties that is pushed through a nozzle and afterwards subjected to processes of drying and stretching, until collected on a grounded support [43]. Briefly, the electrospinning solutions were prepared in two stages, as follows: in the first, the mineral powder was dispersed in the solvent mixture by ultrasonication for 30 min, while in the second, the polymer was dissolved in the obtained suspension by magnetic stirring for 24 h. For each type of powder (D, A, M), two suspensions were achieved, with 5 or 10 wt% inorganic content; PCL concentration was 16 wt% for all cases. The final samples were coded as PCL-D-5%, PCL-D-10%, PCL-A-5%, PCL-A-10%, PCL-M-5% and PCL-M-10%, as a function of loading type and concentration (Table 1).

### 2.3. Investigation Techniques

#### 2.3.1. Physico-Chemical Characterization

The morphology was evaluated by scanning electron microscopy coupled with energy-dispersive X-ray spectroscopy (SEM+EDX) with a FEI Quanta Inspect F electron microscope (FEI Company, Hillsboro, OR, USA), 20 or 30 kV accelerating voltage, 10 mm working distance, and gold coating by DC magnetron sputtering for 40 s. The vibrational characteristics were investigated by attenuated total reflection Fourier-transform infrared spectroscopy (ATR-FTIR) with a Thermo Scientific Nicolet iS50 spectrophotometer (Thermo Fisher Scientific, Waltham, MA, USA), 400–4000 cm^−1^ wavenumber range, 4 cm^−1^ resolution, and 64 scans/sample.

#### 2.3.2. Biological Evaluation

The powder bioactivity was assessed by immersion in simulated body fluid (SBF) prepared according to Kokubo et al. [44], *pH* = 7.3, at 37 °C, for 4 weeks. It is widely accepted that such in vitro studies represent a standard approach to evaluate the apatite-forming capability of implantable materials, as a first step towards hard tissue bonding through a chemically stable and mechanically appropriate interface; by immersion in SBF solutions that mimic the composition of human plasma and selecting suitable testing parameters, reliable data on specimen bioactivity can be achieved [45]. In our case, the solid to liquid ratio was 1:10.

Mouse fibroblasts (L929 cells) were grown in MEM supplemented with 2 mM L-glutamine, 10% fetal calf serum (FCS), 100 units/mL of penicillin, and 100 µg/mL of streptomycin, at 37 °C, in a humidified incubator, under an atmosphere containing 5% CO_2_. All cell cultivation media and reagents were purchased from Biochrom AG (Berlin, Germany).

Cell viability was evaluated using 3-(4,5-dimethylthiazol-2-yl)-2,5-diphenyltetrazolium bromide (MTT) assay at 24 and 48 h after the cells were seeded onto the investigated fibrous composites. Briefly, the surfaces were sterilized in flow with UV light, 15 min on each side. Following the sterilization, 1 cm^2^ squares were placed in 24-well plates and seeded with 20.000 cell/well for 24 and 48 h. After the desired time passed, the medium was exchanged with medium containing 1 mg/mL MTT and further incubated for 4 h in the incubator. Finally, the solution was extracted and the formed formazan crystals were dissolved in dimethyl sulfoxide (DMSO). Negative control was represented by cells cultivated on aluminum foil. The percentage of viable cells was obtained using Equation (1).
*Cell viability = [(A_570_ of treated cells)/(A_570_ of untreated cells)] × 100    (%),*(1)

Morphological investigation was performed using fluorescence microscopy for cells grown for 24 h on the developed scaffolds. The cells were grown in the same way presented above; after 24 h, they were washed with PBS, fixed in 4% paraformaldehyde dissolved in PBS for 15 min, followed by a washing step. Sequentially, the cells were stained for 15 min with 20 μg/mL Acridine Orange solution and washed again with PBS. Finally, the fluorescence images were taken using a confocal microscope (Andor DSD2 Confocal Unit, Belfast, UK) mounted on an Olympus BX51 epifluorescence microscope, employing a 40× objective. The images were recorded using a suitable filter cube (excitation filter 466/40 nm, dichroic mirror 488 nm, and emission filter 525/54 nm).

## 3. Results and Discussion

Figure 1 displays SEM images at different magnifications of the pristine and D-loaded PCL fibers. Figure 1a,b indicates non-woven fibers, randomly arranged in several layers, with an average diameter of 3 μm for the polymeric fibers. The length of the fibers cannot be estimated, but it can be stated that they have a great tendency to gather tightly, sometimes even to stick. Their surface is smooth, the diameter relatively constant along the entire length, and the overall aspect is slightly wavy/winding, suggesting high flexibility. Very rarely, fibers with a much smaller diameter or areas with electrospinning defects can be detected.

The addition of mineral powders (D, A, M) at a proportion of 5 or 10% triggers significant modifications in the general appearance, reducing the fiber diameter to a large extent, below 100 nm, most of them being around 30 nm (Figure 1c,d and Figure 2). This behavior highlights the major changes induced by the presence of the powder in the precursor solution. In addition, the degree of interconnection increases, which makes the scaffolds appear in the form of networks with many points of connectivity, the most congested points being supported by the inorganic aggregates. The presence of micrometric entities homogeneously distributed in the volume of the fiber network is not equivalent to the presence of ceramic aggregates, thus indicating the emergence of polymeric beads on the primary fiber. However, a closer look reveals the existence of several bright areas corresponding to the ceramic bodies. The increase in the loading concentration does not result in significant differences to the general look, which suggests that a large proportion of the powder has sedimented in the precursor solution. The only major dissimilarity appears in the case of D, for which 10% concentration was not favorable for the electrospinning process, resulting in an electrosprayed sample (Figure 1e,f). This is probably correlated with the fact that the powder was thermally processed at a lower temperature (1000 °C) and consists of smaller entities that affect the solution viscosity and rheological properties, hindering the flow process and jet stretching.

Bafandeha et al. [46] fabricated poly (vinyl alcohol)/chitosan/akermanite scaffolds by electrospinning and reported fiber diameters of less than 100 nm, without beads, for 1 wt% akermanite content and with a considerable number of beads for 2 wt% akermanite, which was attributed to the improper viscosity of the electrospinning solution. In our case, 5 or 10% silicate powder was integrated in the precursor system, leading to considerable fiber thinning, of 30 times, as well as large and numerous local diameter enlargements because of the conductivity and viscosity changes occurring after mineral phase addition. An obvious reduction in fiber diameter was also observed for other PCL scaffolds loaded with mineral powders, such as ZnO, TiO_2_, and HAp [47].

Otherwise, the inorganic aggregates are caught as if in a spider web, either covered with a small layer of PCL or just attached to the fiber surface (Figure 2), making their outer side available and prone to biomineralization. Thus, the mineral blocks that are coated with polymer will require a longer time for the bioactivity to be displayed, namely the period necessary for polymer degradation and ceramic surface exposure.

Figure 3 integrates the EDX spectra associated with the simple and composite fibers at the concentration of 5% ceramic powder. If, in the case of PCL, the main elements of the polymer (C, O) are visible, in the other three spectra, additional elements specific to the loading composition can be found (Si, Ca, Mg). In this way, the distribution of the mineral aggregates in the polymeric fibrous network is confirmed.

The loading procedure was also validated from the FTIR spectra (Figure 4) recorded for the pristine and powder-containing fibers. According to the scientific literature, the polymer fingerprint is defined by the vibrational bands typical of C–H, C=O, and C–O, occurring at around 2945, 1720, and 1165 cm^−1^, respectively [48,49]. For the composite fibers, below 1100 cm^−1^, some additional bands emerge; in the range 800–1100 cm^−1^, the signals of Si–O bonds and Si–O–Si groups are visible, whereas below 550 cm^−1^, bands specific to the Ca–O and Mg–O oxide bonds overlap [50,51,52].

In the SEM images taken of the thermally treated powders immersed in SBF for 4 weeks (Figure 5), the behavior of D, A and M during the biomineralization process is noticeable. In the case of D and A compositions, a new morphology is highlighted from place to place, namely quasi-spherical, porous structures, similar to a ball of fibers. As Ca concentration increases from D to A, the spheres grow in diameter (from around 100 nm to about 500 nm) and acquire a better-defined morphology, which suggests greater bioactivity. In the case of M, the morphology is different from the other two, and is also specific to apatite, a tangle of fibrillar structures that completely covers the block surface, of 10 nm in diameter for an individual entity. It can be concluded that the increase in Ca content leads to improved bioactivity, speeding up the healing process and subsequently the implant osseointegration. Furthermore, this in vitro assay validates the multifunctional character of the loaded scaffolds, since the bioresorbability of PCL is complemented by the bioactivity of the silicate powders, opening the possibility for the emergence of synergistic effects. Shahrouzifar et al. [23] obtained diopside scaffolds by sintering at 1200 °C and observed some spare nanometric plates of apatite on the surface of the pure scaffold, while F- and Sr-doped scaffolds were generously covered with apatite microspheres after 1 week of immersion in SBF; all these structures displayed a leaf-like morphology. Since our D powder was thermally treated at 1000 °C and the soaking time was longer (1 week vs. 4 weeks), better mineralization was achieved for an undoped material, confirming that the crystallization degree and grain growth represent determining parameters in the process of mineralization.

When combined with chitosan for the production of scaffolds by 3D-bioprinting, akermanite conferred bioactivity, the surface becoming almost wholly coated with a thick layer of bone-like apatite after 10 days of immersion in SBF; the newly formed precipitate had a sponge-like morphology [10]. Moreover, the soaking period and akermanite concentration were strongly correlated with the degree of coverage, resulting in a dense and homogeneous apatite deposition for 80 wt% akermanite. Thus, it is desirable to have as much akermanite as possible in the fibrous scaffolds, so that a fast and extended mineralization can occur, which will further promote the intimate connection between the living bone and artificial implant, ensuring a stable and lasting bond.

Merwinite has been studied to a lesser extent, but the available data suggest that the rate of apatite growth is extremely high for merwinite compared to akermanite and diopside, following the same trend as CaO content; as observed, the freshly emerged layer displayed a cauliflower-like morphology and completely covered the scaffold surface, the density was extremely favored in the case of calcium resources provided by merwinite [9].

After testing the biocompatibility of the developed composite materials, it was found out that they were not cytotoxic and ensure a favorable platform for fibroblast proliferation. Thus, Figure 6 presents the calculated cell viability, which indicates that a concentration of 10% inorganic powder ensures a slightly better response compared to 5% loading. Moreover, a higher concentration of mineral loading is correlated with a small increase in cell viability with the seeding time.

Indeed, PCL stands out as the sample with the highest cell viability; this result can be explained based on its different morphology compared to all other specimens. In other words, if PCL fibers have a diameter around 3 μm, the size of the powder-containing fibers drops below 100 nm (Figure 1 and Figure 2). Due to its dimensional compatibility at the micrometric scale, the PCL scaffold seems to ensure a better environment for cell adhesion and proliferation. However, the addition of a silicate powder offers an indisputable advantage in terms of bioactivity, which makes this slight decrease in cell viability a compromise we can afford.

The morphological evaluation of L929 cells grown on the investigated materials was performed using fluorescence microscopy; the corresponding images are reported in Figure 7. Similar to the MTT results, the microscopy images reveal the biocompatibility of the electrospun samples, irrespective of the loading type and concentration. The fibroblast cells grown on the control surface show an elongated morphology and an intact oval nucleus, with a normal shape and size (Figure 7A); regardless of the experimental conditions, cell morphology was not altered (Figure 7B–H).

The in vitro assays performed in this paper are in concordance with previous studies, which have confirmed that polycaprolactone-based scaffolds, membranes, or gels have good biocompatibility with osteoblasts, bone marrow mesenchymal stem cells, dental pulp cells, or fibroblast-like cells [49,53,54,55,56,57,58,59,60]. The encouraging results obtained for L929 fibroblasts justify additional investigations on the proposed materials. Further studies should focus on more information regarding cell adhesion and migration, as well as a long-term evaluation of cell behavior in the presence of the samples.

## 4. Conclusions

Calcium has a key role in bone remodeling by modulating the cellular responses to bioceramics, as well as promoting cell growth and osteogenic differentiation. Therefore, three calcium magnesium silicate compositions, corresponding to diopside, akermanite, and merwinite, were processed as thermally treated sol–gel-derived powders and subsequently integrated in polycaprolactone fibers. The fibrous composites were fabricated by electrospinning and evaluated in terms of powder loading degree and particle distribution in the fibers, as well as response to immersion in simulated biological fluid and influence on fibroblast cells. If the inorganic component proved to be bioactive, gradually increasing from diopside to akermanite and then to merwinite (the bioactivity increased with the increase in Ca amount), the loaded fibers represent cell-friendly supports, achieving better cell proliferation when combined with a higher powder concentration.

## Figures and Tables

**Figure 1 polymers-14-04611-f001:**
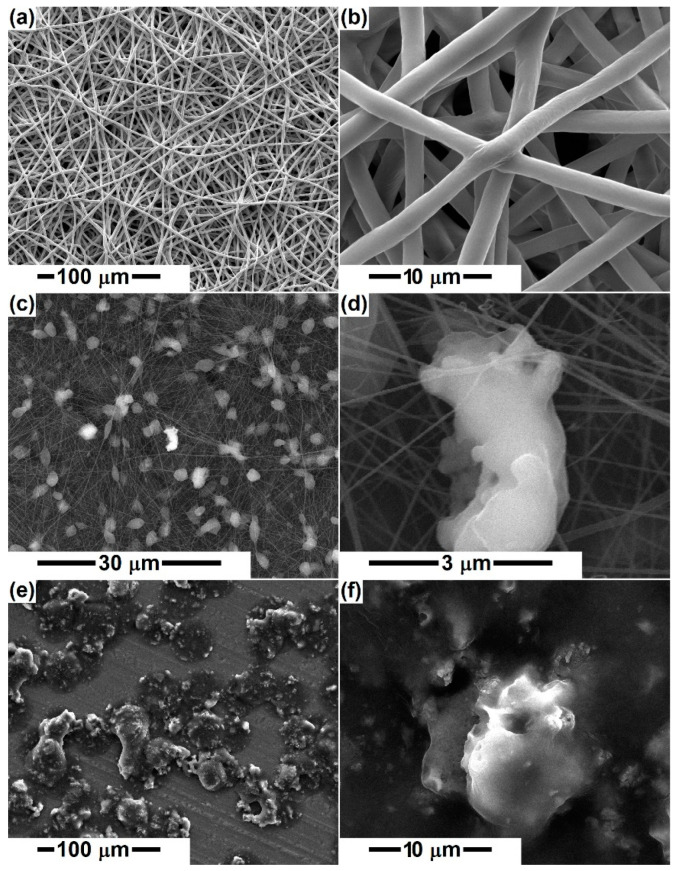
SEM images of the electrospun samples: (**a**,**b**) PCL, (**c**,**d**) PCL-D-5%, and (**e**,**f**) PCL-D-10%.

**Figure 2 polymers-14-04611-f002:**
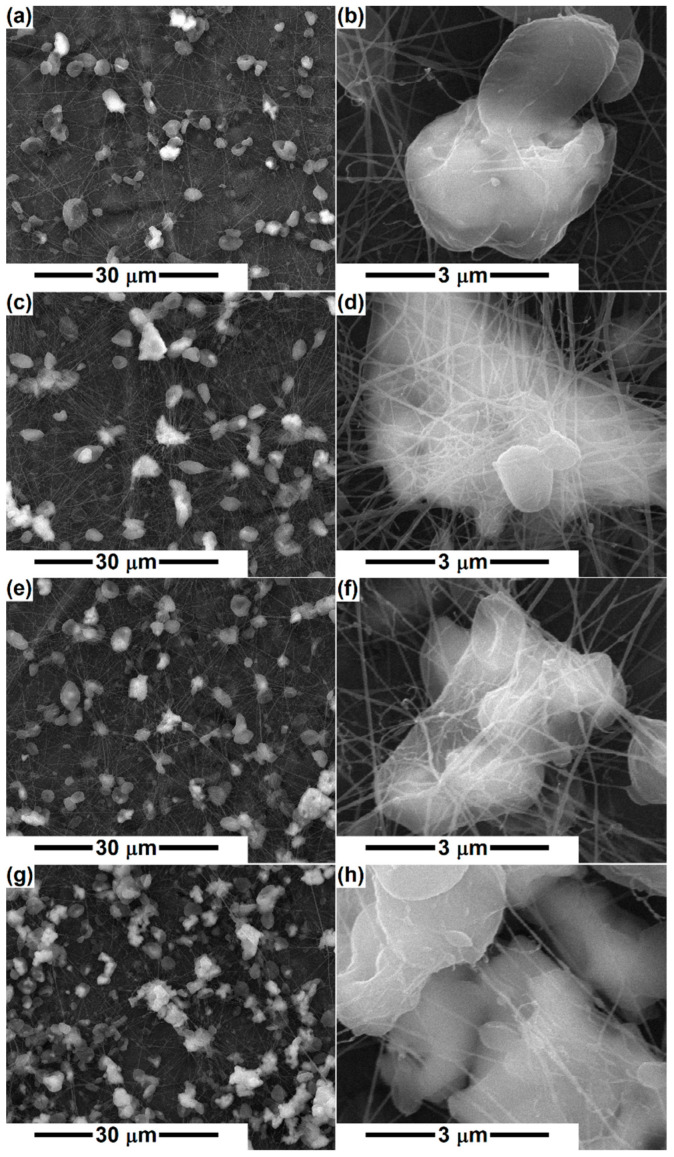
SEM images of the electrospun samples: (**a**,**b**) PCL-A-5%, (**c**,**d**) PCL-A-10%, (**e**,**f**) PCL-M-5%, and (**g**,**h**) PCL-M-10%.

**Figure 3 polymers-14-04611-f003:**
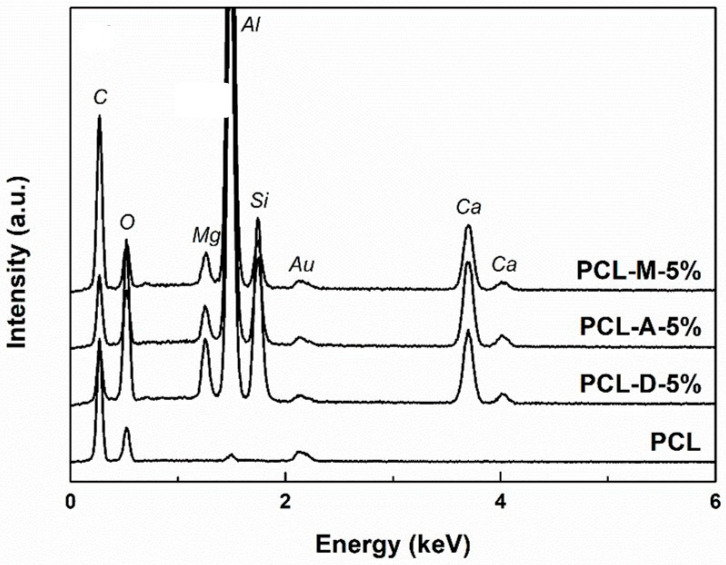
EDX spectra of the electrospun samples with 5% powder loading.

**Figure 4 polymers-14-04611-f004:**
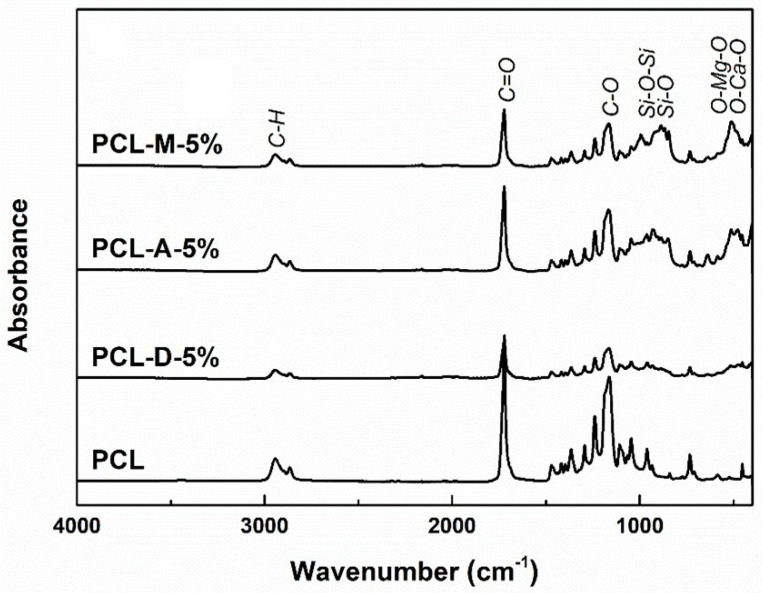
FTIR spectra of the electrospun samples with 5% powder loading.

**Figure 5 polymers-14-04611-f005:**
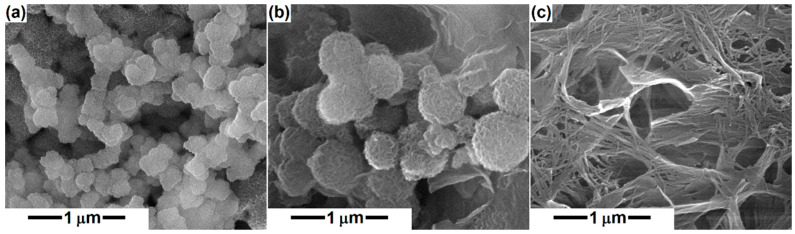
SEM images of the mineral powders after immersion in simulated body fluid for 4 weeks: (**a**) diopside, (**b**) akermanite, and (**c**) merwinite compositions.

**Figure 6 polymers-14-04611-f006:**
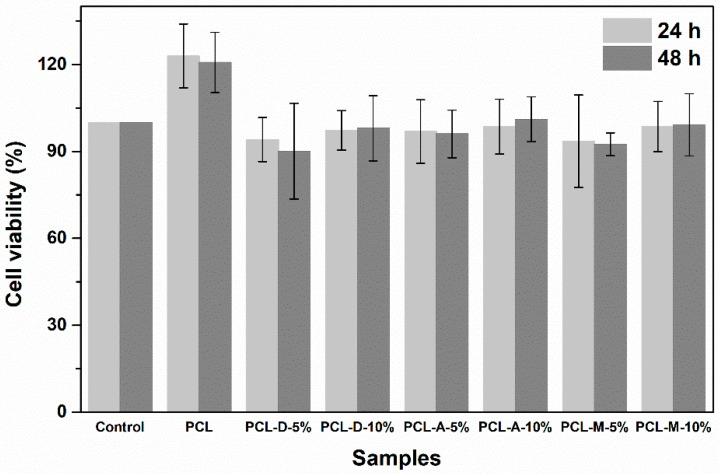
Cell viability for fibroblasts in contact with the electrospun samples for 24 and 48 h.

**Figure 7 polymers-14-04611-f007:**
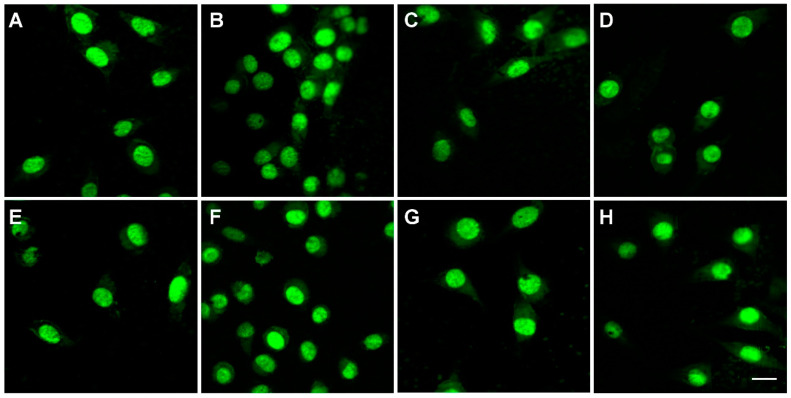
Morphology of fibroblasts in contact with the electrospun samples for 24 h: (**A**) Control, (**B**) PCL, (**C**) PCL-D-5%, (**D**) PCL-D-10%, (**E**) PCL-A-5%, (**F**) PCL-A-10%, (**G**) PCL-M-5%, and (**H**) PCL-M-10%.

**Table 1 polymers-14-04611-t001:** Composition of the electrospinning solutions.

No.	Sample Code	PCL (%)	D (%)	A (%)	M (%)
1	PCL	16	-	-	-
2	PCL-D-5%		5	-	-
3	PCL-D-10%		10	-	-
4	PCL-A-5%		-	5	-
5	PCL-A-10%		-	10	-
6	PCL-M-5%		-	-	5
7	PCL-M-10%		-	-	10

## Data Availability

Not applicable.

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
