# Peer review of "Composite Fibers Based on Polycaprolactone and Calcium Magnesium Silicate Powders for Tissue Engineering Applications"

_polymers, 2022, doi:10.3390/polym14214611_

Round 1

Reviewer 1 Report

The manuscript "Multifunctional Composite Fibers Based on Polycaprolactone  and Silicates Powders for Tissue Engineering Applications" by I.M. Deleanu et al. reports on the preparation and characterization of composite materials based on polycaprolactone fibers reinforced with inorganic powders (calcium magnesium silicates) designed for tissue engineering applications. Overall, the paper is well designed and the data presented here are of high interests in this field.

I identified only minor issues to be addressed:

- in Title, as well as in the Abstract the authors mentioned about multifunctional properties of these composites, that are nor clear highligted within the results and discussion section.

- In Introduction section, at lines 34-36, the authors must add the main classes of polymers used in tissue engineering field to highlight their choice, PCL, respectively. Also they need to include some literature data referring to the mineral powders used for such applications.

- Materials and Methods section requires a reorganization in Materials used to prepare the composites, Preparation of the composite materials and Investigation techniques. A table instead the lines 100-108 would greatly facilitate the understanding of the composition of the materials.

- The Results and discussion Section should contain the data in the order of presentation of the characterization equipment.

- In Fig. 6 how to explain the viability cell higher than 100% of PCL?

Based on these observations, I recommend the acceptance of this paper after Minor revision.

Author Response

On behalf of all authors I want to express our gratitude for your valuable comments and suggestions that helped us to improve our manuscript.

Reviewer 2 Report

Busuioc et al. present an interesting and well-characterized manuscript. However, the authors must make important changes:

-First, the authors should reconsider the title of the manuscript. Authors need to be more specific.

-The summary presented by the authors is inadequate. The authors must rewrite it completely. The authors must adequately introduce the study and the results, with clear conclusions and a translational character.

-The introduction to the state of the art should be improved. Authors may not use 40 references in this section. The authors must be more specific to justify the manuscript. In this sense, the authors must include references with a character of application and translation of these materials.

-The methodology is limited, it should be improved and introduce more references. Authors should specifically improve the Materials characterization section.

-Reference 42 should be reconsidered, and the explanation of this section improved.

-The results are well presented. However, the figure legends of all figures must be upgraded. It is currently a very limiting point.

-The authors must carry out an adequate discussion. In the current version, there is no discussion. This point makes the conclusions unsupported. The authors must make a discussion section and with a translational character.

-The authors must improve very extensively the use of English grammar.

Author Response

(The authors gave the same response as above.)

Round 2

Reviewer 2 Report

The authors have correctly and accurately made all the required changes. The manuscript can be published.

Please review the use of English grammar.